# Improved Therapeutic Efficiency against Obesity through Transdermal Drug Delivery Using Microneedle Arrays

**DOI:** 10.3390/pharmaceutics13060827

**Published:** 2021-06-02

**Authors:** Yixuan Xie, Ruomei Shao, Yali Lin, Chunnan Wang, Ying Tan, Weidong Xie, Shuqing Sun

**Affiliations:** 1Institute of Biopharmaceutical and Health Engineering, Shenzhen International Graduate School, Tsinghua University, Shenzhen 518055, China; xieyx18@mails.tsinghua.edu.cn (Y.X.); srm20@mails.tsinghua.edu.cn (R.S.); lin-yl19@mails.tsinghua.edu.cn (Y.L.); wangcn18@mails.tsinghua.edu.cn (C.W.); tan.ying@sz.tsinghua.edu.cn (Y.T.); xiewd@sz.tsinghua.edu.cn (W.X.); 2Department of Biomedical Engineering, Tsinghua University, Beijing 100084, China; 3Department of Physics, Tsinghua University, Beijing 100084, China

**Keywords:** CL-316243, obesity, degradable microneedle, drug delivery

## Abstract

In this paper, we prepared patches that were composed of a degradable microneedle (MN) array with a soft backing provided for the skin tissue. We then performed a transdermal delivery of anti-obesity drugs to evaluate the effectiveness of β3 adrenergic receptor CL316243 in obesity treatment in overweight mice induced by a high-fat diet. Eighty male National Institutes of Health (NIH) mice were randomly divided into four obese groups or the control group. The obesity groups were given a high-fat diet for 15–18 weeks to establish an obese model. Afterward, the obese groups were divided into the following four groups: the control group, the unloaded MN group, the CL-316243 MN group, and the injection group. For the injection group, the group of mice was injected subcutaneously with CL316243 (1 mg/(kg·day)) for 15 days. Furthermore, the CL-316243 MN group was given a lower dose (0.1 mg/(kg·day)) for 15 days. After weighing the mice, we used Western blotting to detect the expression of uncoupling protein 1 (UCP1) in the adipose tissue around the mouse viscera. The results stated that the weight of the CL-316243 MN group and the injection group dropped, and the UCP1 protein expression of brown adipose tissue (BAT) significantly increased. The results demonstrated the β3 adrenergic receptor agonist CL316243 could be carried into the body through MN, and the dose applied was considerably smaller than the injection dose. The reason for this may arise from the CL-316243 being delivered by MN arrays to subcutaneous adipose tissue more efficiently, with an even distribution, compared to that of the injection dose. This technique provides a new and feasible way to treat obesity more effectively.

## 1. Introduction

Overweight and obesity are major risk factors for many chronic diseases, including diabetes, cardiovascular disease, and cancer. Once considered a problem only in high-income countries, overweight and obesity are now rising sharply in low- and middle-income countries (especially in urban environments). In 2016, 39% of men and women over 18 were overweight (BMI ≥ 25 kg/m^2^), while 11% of men and 15% of women were obese (BMI ≥ 30 kg/m^2^). As a result, nearly 2 billion adults worldwide are overweight, of which more than one billion adults are obese. Over the last 40 years, both overweight and obesity have shown a significant increase. Therefore, there is a growing demand for obesity management and treatment. Fat in mammals can be divided into white adipose tissue (WAT) and brown adipose tissue (BAT). WAT is mainly distributed in the groin and mesenteric area, and its main function is to store fat [1,2,3]. BAT is mainly distributed in the shoulder, shoulder blade, neck, and other areas. Its main function is to convert the calories in food into energy. From the perspective of energy balance, reducing weight by decreasing calorie production and increasing energy consumption can be used as a means to combat obesity [4]. After discovering the active brown adipose tissue in the human body, people began to explore the regulating effect of the thermogenic function of BAT on the treatment and prevention of obesity-related diseases. In addition to WAT and BAT, there is a third type of beige adipose tissue that deserves attention [5]. The phenotype of this tissue is close to BAT, but it has the functions of white adipose tissue and brown adipose tissue, and can transform between the two. The transition from WAT to a BAT-like phenotype has become an attractive method to combat obesity. It is far better than the few drugs [3] currently approved by the U.S. Food and Drug Administration (FDA) as complementary therapies for weight control and diet therapy, that is, anti-obesity drugs that work by suppressing appetite (such as lorcaserin), or reduce fat absorption from the gastrointestinal tract (orlistat). At present, the better drug, found based on the browning of white adipose tissue, is CL316243. CL316243 is a highly selective β3 adrenergic receptor agonist. Related studies [4,5,6,7,8,9,10] have shown that it can promote the decomposition of visceral white fat in obese mice and activate mitochondrial uncoupling protein 1 (UCP1) which induces the browning of white fat (BAT) [4,9]. However, side effects such as nausea, diarrhea, and even cardiotoxicity have been observed when these drugs are applied to patients through traditional administration methods [5,6]. Although conventional hypodermic needles can be injected directly under the skin, long-term treatment at home is impractical because of pain and possible infection at the injection site due to poor patient compliance. Besides, since obesity is not immediately life-threatening, it is important to make the treatment tolerable, safe and convenient. In recent years, as a new type of transdermal drug delivery system, the emerging microneedle (MN) transdermal drug delivery system can not only overcome the problems of drug toxicity and side effects, but also the problem of the low permeability of traditional transdermal drug delivery methods [11,12,13,14,15,16].

MN transdermal administration usually consists of tens to hundreds of hollow or solid MNs (needle length is about 50~900 μm), and a transdermal patch of 1–2 cm^2^ is attached to the skin and pierces the outermost layer. The stratum corneum allows the drug to be fully absorbed to enter the body to play a role [17]. It has dual advantages over injection and traditional transdermal administration. First, it is fast, convenient, painless, and minimally invasive. More importantly, the patient can terminate the administration in time according to the actual drug demand, thereby avoiding the side effects caused by over-dosing [17,18]. At the same time, the MN transdermal drug delivery system can avoid the other disadvantages of oral drug delivery, such as the potential hazards of inhaled drugs and their auxiliary particles to the lungs, extreme pH of gastric acid, degrading digestive enzymes, and intestinal peristalsis. It also has some obvious advantages of transdermal delivery of proteins [19] and macromolecular drugs. For chronic diseases such as obesity, long-term management is required, and patient compliance is particularly important. After the degradable or soluble MNs are inserted into the skin, they will be completely degraded or dissolved in the skin, thereby realizing the release of the encapsulated drug [13,20,21,22,23]. Therefore, the MNs must be made of biodegradable or water-soluble materials. Compared with the previous traditional MNs, there is no biohazardous sharps waste to be disposed of after use [12,16,21,24,25,26,27,28,29]. Hence, the degradable MN transdermal drug delivery system has broad application prospects in the treatment of obesity.

In this work, we use degradable poly(lactic-co-glycolic acid) (PLGA) (extended release drugs) [30] doped with a small amount of polylactic acid (PLA) (increased mechanical properties) [25] to deliver CL316243 anti-obesity drug (Figure 1), and explore its administration effect on NIH mice [20]. The results show that it is an effective way to treat obesity and is more efficient than hypodermic injection delivery.

## 2. Materials and Methods

### 2.1. Preparation of PDMS MN Master Mold

The Polydimethylsiloxane (PDMS) mold was prepared by casting the PDMS prepolymer (elastomer/curing agent = 10:1 by weight) [26,31,32] onto a customized fine copper master (Taizhou Microchip Pharmaceutical Technology Co. Ltd., Taizhou, China) with an array of 10 × 10 pyramid or Cone tips (650 μm in height and 250 μm in base length). After degassing in the vacuum for 30 min and thermal curation at 65 °C for 2 h, the inversely replicated PDMS master mold was obtained by carefully peeling off from the metal substrate.

### 2.2. Fabrication of Composite Biodegradable MNs

MNs were fabricated by sequentially casting two solutions onto the mold (Figure 2). PLGA would provide primary control over the drug release rate [33], and PLA was added to increase the mechanical strength. The casting solution was made by dissolving 0.45 g PLGA and 0.05 g PLA in 2 mL dioxane (Sigma–Aldrich, St. Louis, MO, USA), then adding a solution of 75 μg CL-316243, and finally mixing them with additional dioxane and deionized water to obtain the final casting solution.

To fabricate the blank MN patches, no CL-316243 was added in the polymer solution, which contained 5% (*w*/*v*) of PLGA and PLA (90 and 10% *w*/*w*) in dioxane and deionized water (95 and 5% *v*/*v*). To fabricate the MN patches containing fluorescein isothiocyanate isomer (FITC) (Σ–Aldrich), 20 mg FITC powder was added into the blank casting solution without CL-316243. Next, 20 μL of the casting solution was applied to the top of the MN mold and placed in a vacuum oven for 5 min to fill the mold. Then, 25 μL dioxane was applied to the top of the mold and then placed in a vacuum oven for 5 min to wash the residual casting solution on the top of the mold into the mold cavities. The loading and washing process was repeated three more times to fill the mold, then the mold was dried in a vacuum oven at 60 °C for 6 h.

After that, the second casting solution, consisting of 18% (*w*/*v*) PVA and 18% (*w*/*v*) sucrose (Σ–Aldrich) in deionized water, was gently applied to the dried PDMS mold surface to form the patch backing. After drying in the chemical hood for 2 h, the mold was placed in a desiccator for 2 days at room temperature (20–25 °C) for complete drying, after which the patch was carefully peeled from the mold and stored in a desiccator until use.

### 2.3. Mechanical Strength Measurement

The mechanical strength of MN was measured by force-displacement analysis using the electronic universal testing machine (MTS SYSTEMS (CHINA) CO.LTD, Shanghai, China). MN was tailored into a 10 × 10 array and immobilized on the bottom steel plate. The initial distance between tips of the MN and the top sensor was set at 0.65 mm. The speed of the top sensor toward the MN was 0.1 mm/min. When the probe was pressed against the MN, the force and displacement applied by the instrument were recorded on the computer until the MN was destroyed.

### 2.4. Mouse Skin Penetration Test

Immediately after sacrificing the neck of the NIH mouse, the hair on the back of the mouse was shaved with a cordless trimmer. The hair was removed with depilatory cream, and then wiped clean with saline. Immediately after removing the hair, the skin was cut off and placed on a clean table with the stratum corneum facing downwards. The subcutaneous fat layer was wiped and the connective tissue was cleaned by a cotton ball moistened with physiological saline. The skin was rinsed repeatedly with normal saline. After drying, the prepared mouse skin penetration (MNP) was pressed with the thumb for 30 s, and the pinhole morphology of the MNs on the skin was observed. For easy observation, we used 0.4% trypan blue solution to immediately stain the pinhole array area. After 1 min, the dye remained on the skin surface was washed and the skin was observed and photographed.

### 2.5. Skin Healing after Microneedling

To test the effect of the homemade MN on the human body, we prepared a blank MN patch and applied it on the inside wrist of a volunteer, removed it after pressing for 30 s, and recorded the healing process in real-time with an electronic magnifying glass.

### 2.6. Animals

NIH mice were cultivated by the National Institutes of Health. They were widely used in pharmacological virulence pathology and bacteriology research, as well as in the identification of biological products and drugs. 4-week-old NIH mice (level, SPF, male) were used as the experimental objects and were raised to 15 to 18 weeks old to construct the mouse obesity model (DIO). Four-week-old male NIH mice were obtained from Guangdong Medical Animal Center (Guangdong, China). Animals were kept in an environmentally controlled breeding room (temperature: 20 ± 2 °C; humidity: 60% ± 5%; 12 h dark/light cycle). The animals were fed a standard laboratory chow diet (Guangdong Medical Animal Center) with water ad libitum and fasted from 9:00 am to 3:00 pm before the experiments. The study was performed in strict accordance with the recommendations of the Guide for the Care and Use of Laboratory Animals of the Institutional Animal Care and Use Committee of Tsinghua University. The protocol was approved by the Bioethics Committee of Shenzhen International Graduate School, Tsinghua University, China (the project identification code is: Ethics issue [2021] No. 37, 20 January 2020). The constructed mouse model was divided into four groups, namely, control group, unloaded MN group, CL-316243 MN group, and injection group. These four groups were designed to explore the effect of transdermal drug delivery of polymer MNs.

### 2.7. Histology

Mouse tissues were fixed and embedded in paraffin (for hematoxylin-eosin (H&E) staining) or frozen (for oil red staining). Sections were stained with hematoxylin and eosin or oil red according to standard protocols. Specific skin paraffin section procedures and precautions were as follows:Extraction: Fresh tissue was fixed at 4% paraformaldehyde for more than 24 h. The tissue was taken out from the fixation solution and used a scalpel to smooth the tissue at the target site in the fume hood. The cut tissue and the corresponding label were placed in the dehydration box;Dehydration: The dehydration box was put into the hanging basket and dewatered by the gradient alcohol in order: 75% alcohol 4 h, 85% alcohol 2 h, 90% alcohol 2 h, 95% alcohol 1 h, absolute ethanol I 30 min, anhydrous ethanol II 30 min, alcohol benzene 5–10 min, xylene I 5–10 min, xylene II 5–10 min, wax I 1 h, wax II 1 h and wax III 1 h;Embedding: The tissue was soaked in the embedding machine. The melted wax was first put into the embedding frame. Before the wax solidified, the tissue was removed from the dehydration box and put into the embedding frame according to the requirements of the embedding surface, and attached by the corresponding label. It was cooled on a freezer at −20 °C. After the wax solidified, the wax block was removed from the embedding frame and trimmed;Slicing: The trimmed wax block was placed on a paraffin microtome and sliced to a thickness of 4 μm. The slices were floated on the spreader at 40 °C in warm water to flatten the tissue. The tissue was picked up with a glass slide and placed in a 60 °C oven to bake slices. After the water-dried wax was roasted, it was taken out at room temperature and stored for later use;Paraffin section dewaxing to water: The sections were put into xylene I for 20 min, xylene II for 20 min, absolute ethanol I for 10 min, anhydrous ethanol II for 10 min, 95% alcohol for 5 min, 90% alcohol for 5 min, 80% alcohol for 5 min, 70% alcohol for 5 min and finally distilled water;Hematoxylin-stained cell nuclei: The sections were Harris hematoxylin-stained for 3–8 min, washed with tap water, differentiated with 1% hydrochloric acid alcohol for several seconds. They were rinsed with tap water, followed by 0.6% ammonia water returned to blue and finally rinsed with running water;Eosin-stained cytoplasm: They were immersed in eosin staining solution for 1–3 min;Dehydrated slides: The sections were placed in order of 95% alcohol I 5 min, 95% alcohol II 5 min, absolute ethanol I 5 min, anhydrous ethanol II 5 min, xylene I 5 min, xylene II 5 min. The sections were dehydrated and transparent. They were taken out from xylene, dried slightly, and sealed with neutral gum;Microscopy: image acquisition and analysis.

### 2.8. ELISA Analysis

After 10 days of the experiment, the mice were fasted for 12 h, and the blood was collected through the orbit of the glass tube. The collected blood was placed at room temperature for 2 h and then centrifuged at 1000× *g* for 20 min and stored at −80 °C. The triiodothyronine (T3) and thyroid stimulating hormone (TSH) kits were used to determine the content of T3 and TSH in serum by competitive inhibition ELISA (CLOUD-CLONE CORP, Houston, TX, USA).

### 2.9. Western Analysis

The blank control group, unloaded MN group, CL-316243 group, and injection group were treated for 48 h of adipocytes. The total fat protein was extracted, and the protein concentration was measured by bicinchonininc acid (BCA) method, while the UCP1 protein was detected by Western blot. A total of 35.43 μg of protein was loaded on each lane on the SDS-PAGE gel, electrophoresis, transfer, and blocking. The primary antibody was diluted 1000-fold and incubated overnight at 4 °C with a shaker. The membrane was washed with TBST for 5 min × 3 times, and the secondary antibody labelled with Horseradish Peroxidase (HRP) was added (1000-fold dilution). It was incubated for 1 h. After washing 3 times, the color was developed by 3,3′-diaminobenzidine (DAB). After that, it was scanned for characterization.

### 2.10. State Analysis

GraphPad Prism software (version 8.2.0 (272), 16 July 2019, Inc., La Jolla, CA, USA) was used to process the data and to perform statistical significance. The error bars indicated standard deviation (SD). Significance analysis was performed by Ordinary one-way ANVOA test and expressed as follows: *, *p* < 0.05; **, *p* < 0.01; ****, *p* < 0.0001.

## 3. Results

### 3.1. Characterization of Composite Dissolving MNs

We used a two-step casting process. In the first step, a mixture of the drug and PLA/PLGA solution was applied to the PDMS (polydimethylsiloxane) mold to form a needle tip. In the second step, PVA/sucrose was applied to the PDMS mold to form a base. The finished patch consisted of 100 (10 × 10) cones with 250 μm diameter at the base, and 650 μm in height (Figure 3B). Another array was composed of 100 (10 × 10) pyramids with 250 μm long at the base, and 650 μm in height (Figure 3C). The distance between the needle tips was 500 μm (Figure 3A–C). Furthermore, the backing of the microneedle fabricated by flexible PVA made it suitable for MN application on irregular skin. The SEM images (Figure 3B,C) showed the smooth surface and sharp tip of the needles, which proved the MN was in good shape.

### 3.2. MN Insertion Study and Drug Loading Performance

Good mechanical property is critical for the microneedles to overcome the skin barrier and realize efficient transdermal drug delivery. As shown in Figure 4A, a mechanical compression test was performed to evaluate the strength of microneedles using an electronic universal testing machine (MTS SYSTEMS (CHINA) CO.LTD). Mechanical tests of two types of MN (pyramids; cones) with a needle length of 650 μm were carried out. The results were shown in Figure 4B. Before the extreme pressure, the relationship between displacement and force was almost linear. Once the microneedle was interrupted, the curve shook suddenly, and the arrow in the figure indicated its shaking point. As shown in Figure 4B, once the compressive force of each needle exceeded 0.2 *n*, the conical microneedle was broken, and when it exceeded 0.3 *n*, the pyramidal microneedle was broken. According to previous studies [34], a microneedle with a tip diameter of 24 μm required a force of 150 mN/needle to penetrate human skin. As for the needle diameter, microneedles with sharper needles were usually easier to insert into the skin than blunt ones. The tip of this microneedle was 10 μm, and even if it was a conical microneedle with a slightly lower hardness, it could still withstand 0.2 *n* in a single root, which means that its mechanical strength was far enough to penetrate the skin. After the penetration experiment, as shown in Figure 4C, the pinhole array on the surface of the mouse skin could be seen. After application to NIH skin (we quickly pressed a 10 × 10 needles with a length of 650 μm MN into the depilated skin and kept for 30 s), micropores could be observed at the insertion site of the skin (Figure 4D) and the pinhole array on the skin was visible. The shape was just the same as that of the penetration patch. The viable epidermis selectively stained with 0.4% trypan blue solution thus demonstrated the puncture marks of MN (Figure 4E). Additionally, the cross-section stained by H&E demonstrated the microchannels were caused by MN (Figure 4F). MN penetrated to a depth of approximately 270 μm, which was about half of the actual needle length of MN due to the skin elastic property and MN deformation during insertion. All these results confirmed the effective insertion. The degradability of PLGA and PLA enabled almost complete degradation of MN in the skin and the residual base of needles was shown in Figure 4G. This indicated that approximately 75 μg CL-316243 was delivered into the skin. The MN was removed after 30 s of pressing on the inner wrist of the human body, and the shot was taken every two minutes. It could be seen that the inserted skin quickly recovered and completely recovered after 6 min, showing that the micro-targeted skin hardly produces redness and inflammation (Figure 4H).

The CL-316243 solution was used as the drug solution. The morphology under the microscope was shown in Figure 4I,J. Under a confocal microscope, the loaded drug solution with FITC was distributed very evenly throughout the MN needle structure, confirming the feasibility of the method-prepared micro-targeting solution-loaded drug (CL-316243).

### 3.3. In Vivo Studies on Diet-Induced Obese (DIO) Mice

When the mice in the DIO group weighed more than 20% of the control group, the modeling was completed. As shown in Figure 5A, the DIO model was successfully built. 15 days after the administration, the feed intake of the four groups of mice was observed, and their weights were measured. The results showed that there was no significant change in feed intake, but the mice in the CL-316243 MN and Injection groups became lighter, indicating that the metabolism of these two groups of mice was enhanced, while the data in the unloaded and control groups were similar, indicating that this treatment had no effect on the metabolism of mice, and the results were all related to the contained drugs (Figure 5B,C).

Mice were dissected for further investigation. For MN-administered mice, the total BAT organ weight increased (+35.94%), eWAT (epididymal WAT, −42.85%) and iWAT (inguinal WAT, −42.31%) decreased (Figure 6A–C). In addition, the histology of adipose tissue analysis showed a significant reduction in lipid content and adipocyte size in WAT (Figure 6D).

Obese people often find a certain increase in the value of triiodothyronine (T3) or thyroid stimulating hormone (TSH). This is because epinephrine can induce UCP1. As a protein that stimulates thermogenesis, UCP1 can eliminate the mitochondrial proton gradient caused by ATP production. A key factor in the completion of this pathway is type 2 deiodinase, which can increase the content of T3 and TSH. After the administration experiment, the contents of T3 and TSH in the serum were analyzed by ELISA, and the results showed that the content of the CL-316243 group was increased, but it was not very obvious (Figure 6E,F).

A defining attribute of brown or beige adipocytes is their expression of UCP1. UCP1 presents in the inner membrane of mitochondria. This disrupts the proton gradient generated in oxidative phosphorylation by increasing the permeability of the inner mitochondrial membrane. Compared with the control group, the UCP1 protein expression in the CL-316243 MN group is higher (Figure 6G).

## 4. Discussion

Obesity is a risk factor for diseases such as high blood pressure, coronary heart disease, diabetes, and has become one of the most important public health issues threatening human health. In the treatment of obesity, most weight-loss drugs have left clinical application due to various side effects. In recent years, due to its unique physiological functions, brown fat has become an important target for the study of obesity drugs. Many studies have shown that CL316243 played a certain role in the process of inducing WAT browning in mouse experiments with an anti-obesity effect. Obesity is a chronic disease, and long-term injections can cause side effects, and even infections that affect patient compliance, and oral bioavailability is extremely low. Therefore, a new route of administration is needed. In this study, MNs as a new type of transdermal drug delivery system can solve these problems. In all types of MN, biodegradability has many advantages in terms of drug-loading preparation and safety. What is important is that it does not produce any sharp waste. The degradation products of PLGA used in this article are lactic acid and glycolic acid, which are by-products of human metabolic pathways. Therefore, when used in medicine and biological materials, it does not have toxic side effects. Besides, the addition of an appropriate PLA blend can increase its penetration mechanical strength. Overall, MNs made by PLGA are a promising transdermal drug delivery system that slowly releases chronic drugs such as anti-obesity drugs.

In actual applications, the needle of the soluble MN patch needs to be hard enough to pierce the skin, and it is hoped that the patch part has a certain degree of flexibility to closely fit the skin. Considering that the mildness of PLGA-MN may not be enough to penetrate the skin to a certain depth, in this study, an appropriate amount of PLA was added to enhance its mechanical strength. The strength of the cone-shaped MNs tested by the universal testing machine was about 0.3 *n* per needle, which was enough to insert into the skin. Then, we performed insertion studies on NIH mouse skin, because mouse and human skin have a similar structure and are very easy to access. The results showed a penetration depth of approximately 270 µm, which was close to half the height of the MNs. To enhance the flexibility of the backing, PVA/sucrose was used as the backing material and tested on the inside of the human wrist, which showed that it was very skin-friendly. The skin recovered in just 6 min after the removal of MN, indicating that there were no side effects, such as inflammation or redness on the skin. Therefore, we have confirmed that the effective insertion of MN ensures future applications in clinical trial patients.

The drug-loaded MNs prepared in this study produced good results in DIO model mice. Eighty male NIH mice were randomly divided into the control group (20) and the obese group (60). The obesity group was given a high-fat diet for 15–18 weeks to establish an obesity model. Subsequently, the obese group was divided into the control group, the needle-free MN group, the CL-316243 MN group, and the injection group. For the injection group, mice were injected subcutaneously with CL316243 (1 mg/(kg d)) for 15 days, and the CL-316243 MN group was given a low dose (0.1 mg/(kg d)) mice for 15 days. After weighing, Western blotting was used to detect the expression of uncoupling protein 1 (UCP1) in the adipose tissue around the mouse viscera. Compared with the control group, the CL-316243 MN group and the injection group lost weight, while the UCP1 protein expression of BAT increased significantly. The drug affected the browning of white fat. It showed that MN could carry the β3 adrenergic receptor agonist CL316243 to deliver the drug into the body, and the dose used was much smaller than the injection dose. For this chronic disease, it not only solved the inflammation problem of long-term injection, but also saved on the cost of such an expensive drug (1140 RMB per 1 mg). Some clinical studies on humans have shown that after an oral dose of 1500 mg/day of CL-316243 for eight weeks, the bioavailability of CL-316243 was extremely poor due to the first-pass effect of the liver and gastrointestinal effects, making it impossible to further lose weight or reduce fat mass [35,36]. Using the microneedle prepared for administration, and assuming that we standardized the effective dose, as determined in mice, a lower effective dose could be achieved (according to the human weight of 60 kg, it is estimated to be 0.578 mg/day, far lower than 1500 mg/day [36]). The microneedle system has great prospects in later clinical research. The drug dosage can be adjusted by increasing the size of the microneedle patch. Multiple microneedle patches can also be used to achieve this goal [37]. Many in vivo studies have shown that MN has a good effect on health conditions such as diabetes [38], cancer [39], and rheumatoid arthritis (RA) [40]. The traditional injection dose is 10 times that of the MN, but the MN effect is still as significant as the traditional injection. The main reason for this lies the fact in that the traditional injection method leads to a high concentration of medicine, and there is no blood vessel in the fat, so the medicine spreads slowly and the efficiency is relatively low. While polymer MNs can deliver drugs more dispersedly and evenly, the drugs can reach the fat more effectively. MN administration can pierce the epidermis without reaching the dermis, and form micropores in the skin to improve the efficiency of the drug’s transdermal absorption line. These studies confirm the potential of MN. Recently, many groups have explored new manufacturing techniques and have evaluated the application of large-scale MN patches on human volunteers. In our research, MN with high biocompatibility and high bioavailability is easy to manufacture. The principle of the preparation method is simple, which provides opportunities for mass production. In future clinical trials, the patch area needs to be enlarged by 100 times, that is, the side length is increased by 10 times. The original experimental patch has a side length of less than 1 cm. As long as the patch is made into 10 cm × 10 cm, the demand can be met. Such a patch can be applied to the lower abdomen like a plaster without affecting its use. We are convinced that these advances will make the commercialization and clinical application of this type of MN possible.

## 5. Conclusions

In short, the polymer MN patch developed using PDMS molds is very effective for the transdermal delivery of anti-obesity drugs. The MNs prepared with our materials and methods have strong mechanical strength and soft skin fit and can be simply pressed with a thumb. Equipped with CL-316243, the MNs effectively avoid the side effects of oral administration and the high dose (1 mg/(kg·day)) required for injection, reaching 0.1 mg/(kg·day). Later clinical trials can be implemented. In general, we believe that this polymer MN loading drug has broad prospects.

## Figures and Tables

**Figure 1 pharmaceutics-13-00827-f001:**
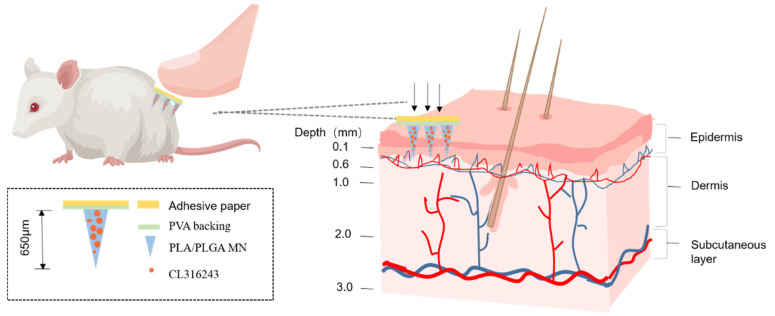
Schematic diagram of the concept. After applying the microneedle (MN) system to the groin on one side of the mouse, CL316243 was released from the MN and absorbed by the tissue capillaries. The dashed frame shows the composition of the microneedle materials.

**Figure 2 pharmaceutics-13-00827-f002:**
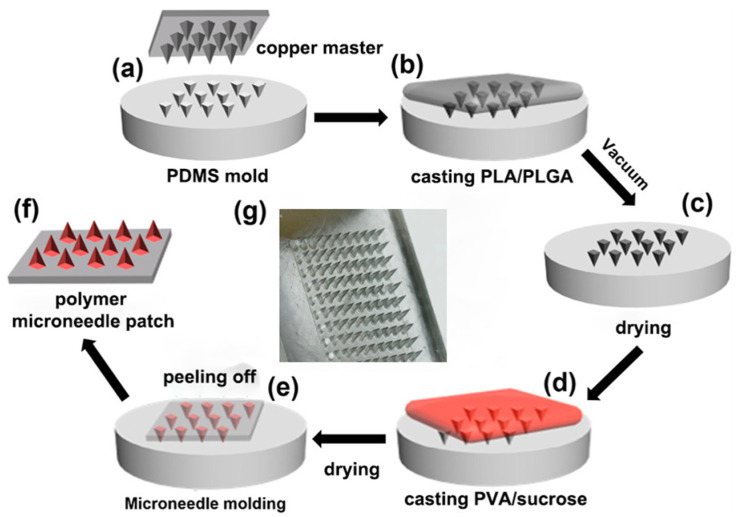
Schematic of the fabrication process of drug-loaded MN-patches: (**a**) replication of PDMS negative molds by casting the prepolymer onto a copper master; (**b**) casting PLA/PLGA onto the PDMS molds with microcavities; (**c**) fabrication of PDMS mold by vacuum; (**d**) casting PVA/sucrose prepolymer onto the PDMS mold; (**e**) drying the mold; (**f**) peeling off the polymer microneedle patch from the PDMS mold; (**g**) the physical picture of the microneedle by electronic magnifier.

**Figure 3 pharmaceutics-13-00827-f003:**
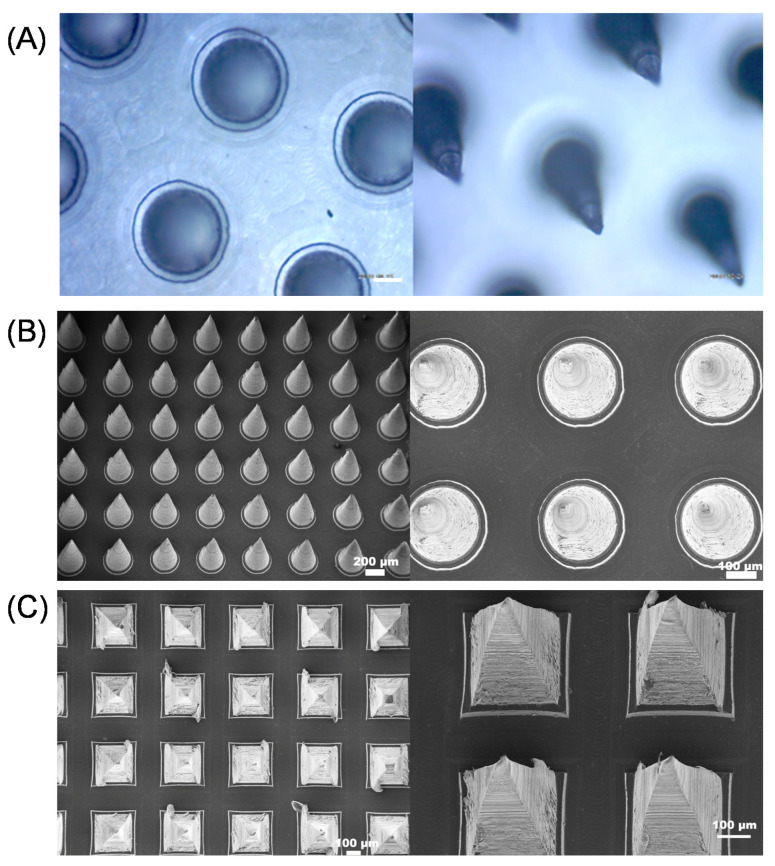
(**A**) Top view under an optical microscope (scale bar: 100 μm) (left) and side view under optical microscope (right); (**B**) scanning electron microscopy (SEM) images of cone MN; (**C**) SEM images of pyramid MN.

**Figure 4 pharmaceutics-13-00827-f004:**
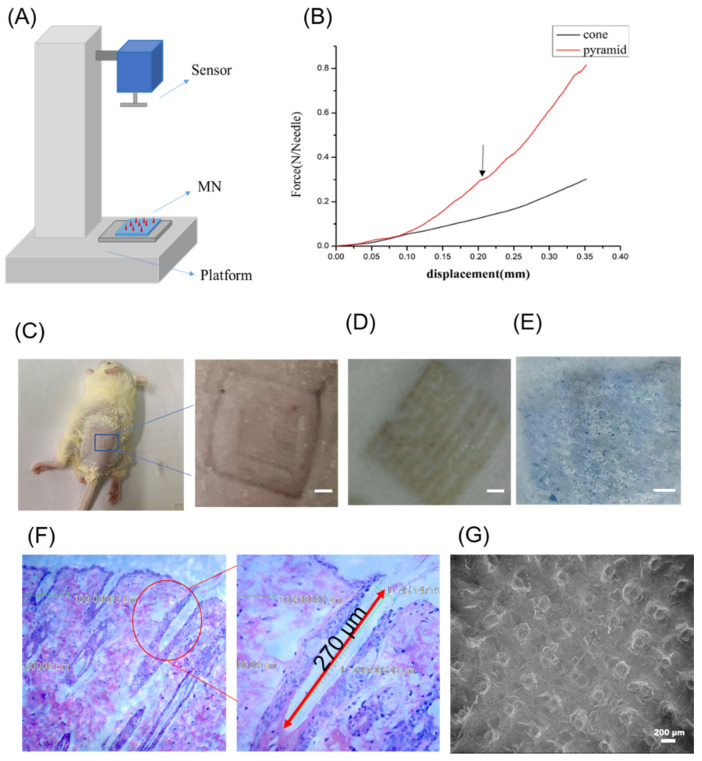
(**A**) Schematic diagram of compression test; (**B**) The force-displacement curves of cone or pyramid MN; (**C**) Pinhole image of mouse micro-skin after 30 s of pressing (scale bar: 2 mm); (**D**) Mouse skin peeled within 1 min after microneedle puncture (scale bar: 2 mm); (**E**) Trypan blue staining image of mouse skin administration (scale bar: 2 mm); (**F**) H&E-stained cross-section of inserted skin by MN; (**G**) SEM image of MN after application to the skin; (**H)** Skin recovery images at 0 min, 2 min, 4 min and 6 min after MN treatment (scale bar: 2 mm); (**I**) Fluorescent confocal stereogram with FITC drug-loaded microneedles (scale bar: 500 μm).; (**J**) Fluorescent confocal top view of FITC drug-loaded microneedles (scale bar: 500 μm).

**Figure 5 pharmaceutics-13-00827-f005:**
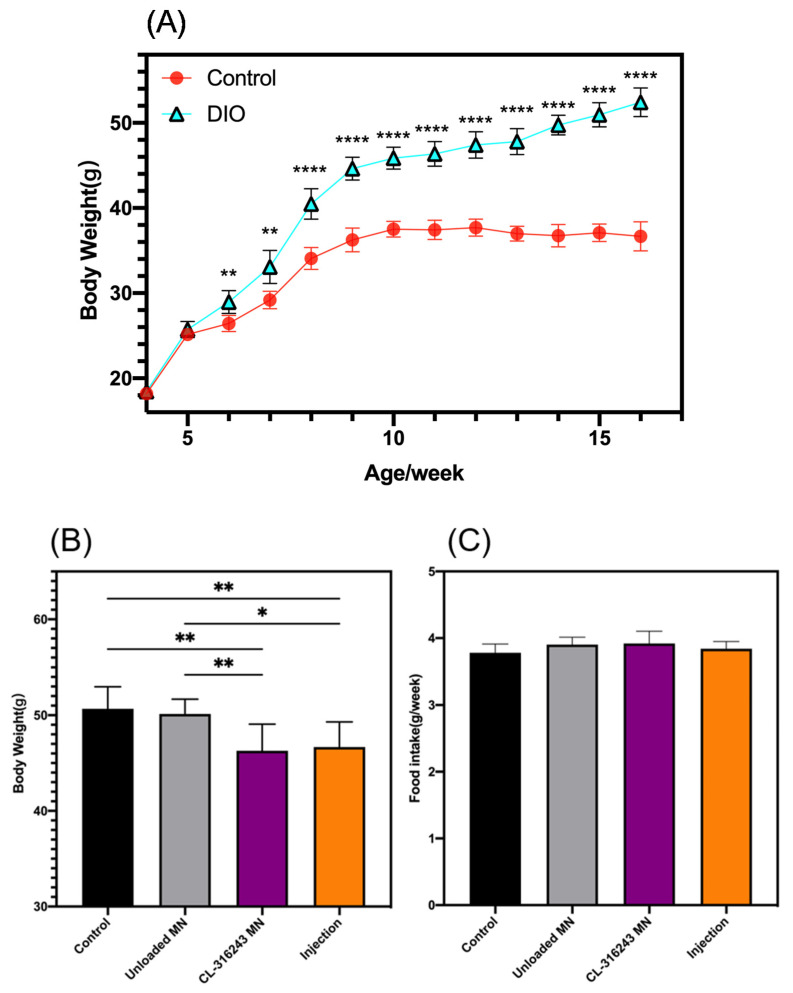
(**A**) Body weight in model group and control group; (**B**) Food intake of mice after dosing; (**C**) Body weight with different treatments. (**A**) Error bars indicate standard deviation (SD), two-tailed Student’s *t*-test, ** *p* < 0.01, **** *p* < 0.0001 compared to control group (*n* = 10). (**B**,**C**) Error bars indicate SD, Ordinary one-way ANVOA test, * *p* < 0.05, ** *p* < 0.01 (*n* = 15).

**Figure 6 pharmaceutics-13-00827-f006:**
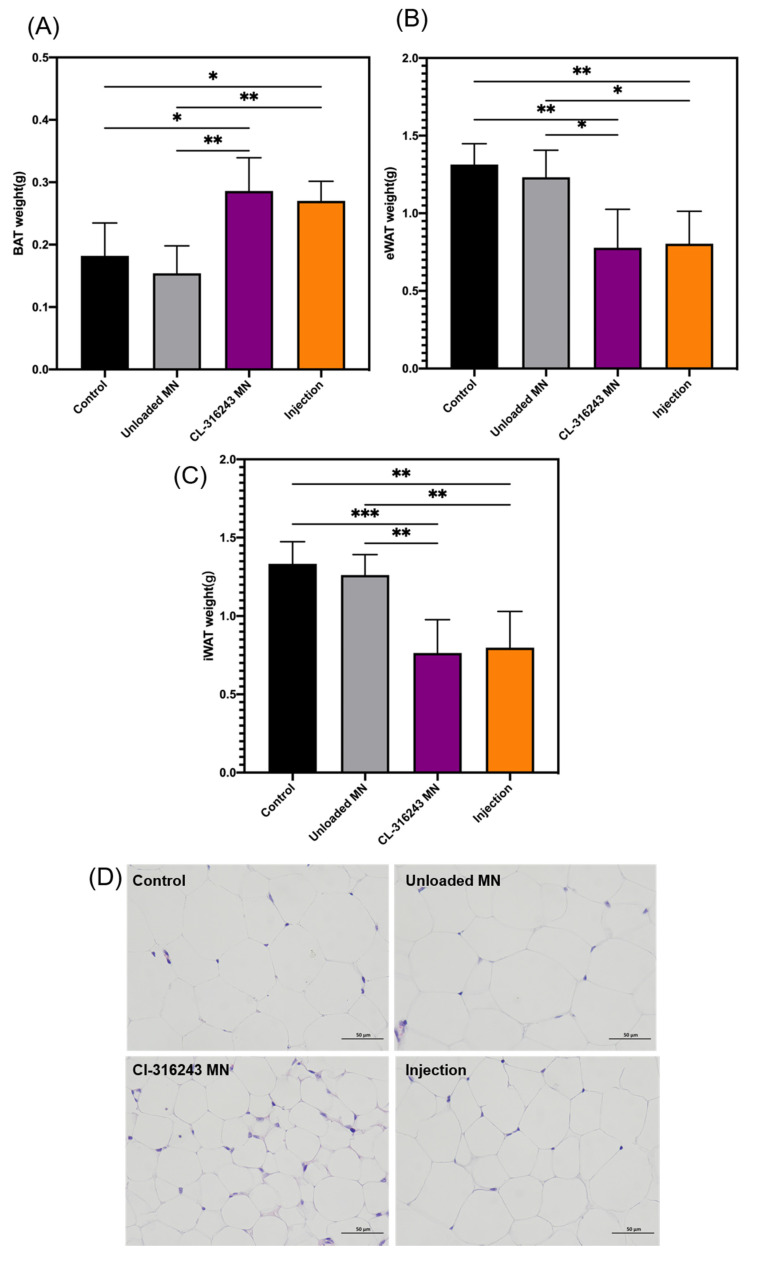
(**A**) BAT weight with different treatments; (**B**) eWAT weight with different treatments; (**C**) iWAT weight with different treatments; (**D**) Representative H&E staining images of WAT (scale bar: 50 μm); T3 (**E**) and TSH (**F**) levels in mouse serum; (**G**) BAT UCP1 protein expression. (**A**–**C**,**E**,**F**) Error bars indicate SD, Ordinary one-way ANVOA test, * *p* < 0.05, ** *p* < 0.01 (*n* = 15).

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
