# Peer review of "Improved Therapeutic Efficiency against Obesity through Transdermal Drug Delivery Using Microneedle Arrays"

_pharmaceutics, 2021, doi:10.3390/pharmaceutics13060827_

Round 1
Reviewer 1 Report
Authors encapsulated CL316243 in microneedle for obesity treatment. In vitro and in vivo studies were performed to certificate efficiency of anti-obesity microneedle patch. However, a few points should be improved for publication.
1. Novelty of this manuscript compared to previous studies using CL316243 as a anti-obesity drug is not clear.
2. Typo errors and grammar of English should be improved.
3. Figure quality is too low and not organized, therefore, content of this manuscript could not be demonstrated clearly.
4. Statistical analyzation is insufficient, especially in figure 5 (b) and (c).
Reviewer 2 Report
The authors of the manuscript titled "Improved therapeutic efficiency against obesity through trans-2 dermal drug delivery using microneedle arrays" are presenting an alternative and interesting approach to the delivery of an anti-obesity drug. There are some points that should be addressed before publication as follows:
It is not clear if on the patch after application there is a residue of microneedles considering the limited penetration depth as compared to the total microneedles length. It should be indicated.
Connected to the previous point, do you have an estimation of the amount of microneedles volume and therefore of drug administered?
In the translation from studied animal to human, considering that there are 2 orders of magnitude of difference in weight how do you think it can be managed the dosage issue?
- Increasing the dosage? If yes how much one could gain per area of patch?
- Increasing the number of patches or patch area? Again how much one can gain?
Some considerations in this direction should be provided in the discussion.
There are some typos in the overall manuscript that should be revised.
Fig. 2 has typos (micronededle). Part C, D and E should be described in the Figure caption. In the Fig.2 On the right side Is the order of a and b correct?
Fig. 4 Scale bar should be more readable and need to be enlarged in Fig.4i.
Fig. 5 and 6 The figure caption should be self explanatory.
Please use more readable fonts such as Helvetica and Arial for the figures. Please modify the figures throughout the manuscript.
There are some highly relevant references should be added in the manuscript.
Recommended literature: 10.3389/fbioe.2019.00296 and 10.1039/d0tb02279a on multicompartment MNs. 10.1002/adtp.202000171 on microneedles for obesity application.
Reviewer 3 Report
In this manuscript the authors compare the efficacy transdermal delivery of anti-obesity drugs to traditional injection methods, an area of high interest to the reviewer. The following changes need to made to enhance the clarity of the findings:
1) The style of the paper needs to be changed to remove adjective such as pristine, successful, prospects. These adjectives need to be measured against standards discussed in the paper.
2) It would be beneficial to create flowchart for the microneedle fabrication process. As it is details are scattered in results and methods. It is not clear what the differences are between the pyramid and conical shape patches. The legends to for Figures 3 have no scale.
3) The mechanical characterization curve in Figure 4 b is not clear neither are the results nor the description of failure. The stress/strain curve for each microneedle type show two brittle compounds so the authors need to describe the failure better. "Once MN breaks, the curve will suddenly shake".
4) The results of the statistical comparison are not well explained there is a single p value for Figure 5 and none for Figure 6. This is very important for the argument of dosage. In addition the number of replicate should also be stated in the legend as well as mentioning whether error bars indicate standard deviation or stand error.
Round 2
Reviewer 1 Report
1.The results section should be written with past tense.
2. Many typo should be correct.
- There are several mistakes to insert space before bracket.
One example is on page 2.
Besides, it has some obvious advantages of 82 transdermal delivery of proteins[19] and macromolecular drugs.
- In page 7, please delete ( after (B),
(B) (scanning electron ----> (B) scanning electron
Figure5 ---->Figure 5
3. Although the author described the penetration of MN into skin, the information of delivered drug is required.
How much drug is actually delivered into skin?
Reviewer 2 Report
Critical points have been addressed. However, some minor points have to be revised.
It has been cited by mistake the following paper: ACS Nano 11 at ref 19 instead of the suggested ref: 10.1039/d0tb02279a
At lines 235-237: “And the distance between the needle tips is 500 μm (Figure 3A-C). Besides, the backing of the microneedle is made by PVA had great flexibility and toughness, which is suitable for MN Application on irregular skin.” There are some grammar mistakes:
- Do not start with End
- There are two verbs “is made” and “had” in the same sentence
- Application should not start with Capital letter
At line 260 “enable” should be replaced with “enables”
In the caption of Figure 4. “Seconds” should be replaced with “s” (somewhere else “minute” should also be replaced with “min” there are also some extra spaces and other word with no spaces in between. Please check them.
Please check also some typos in the caption of Figure 5 and 6.
At line 360 the sentence “making this chronic oral CL-316243 Failure to lose 360 weight and fat mass is not clear.
At line 364 “Much” should be replace with ”much”.
Reviewer 3 Report
Dear Authors,
Thank you for making the suggested changes. The specific aims have greatly improved.
Section 2.10: Do you mean statistical analysis?
Do you have any statistical analysis done around the stress/strain curve?
Section 3.2 needs to be completely re-written. The stress-strain curve is not explained. What is the name of the mechanical test that you ran? You need material terms to describe failure. Putting an arrow on Figure 4B is not an explanation.
The same rigor should apply to the other concepts covered in that section.
Round 3
Reviewer 1 Report
Authors revised manuscript based on reviewer's comment.
Reviewer 3 Report
Thank you for the additional clarifications.